# A comprehensive catalogue of receptor-binding domains in extracellular contractile injection systems

Nimrod Nachmias[1,6], Zhiren Wang[2,3,4,5,6], Xiao Feng[2,3,4,5], Feng Jiang [2,3,4,5] ✉ & Asaf Levy [1] ✉

Extracellular contractile injection systems (eCISs) are bacteriophage tail-derived toxin delivery complexes in prokaryotes. They play roles in microbial interactions with hosts, using tail fiber proteins for target cell binding. Here, we present a comprehensive analysis of eCIS tail fiber genes in bacterial and archaeal genomes, providing insights into their remarkable diversity, target cells, functional adaptations, and evolutionary dynamics. We identified 3445 eCIS tail fiber proteins encoded in 2585 eCIS loci from 1069 microbes. These fibers can be categorized by five new N-terminal domains responsible for tail fiber attachment to eCIS baseplates. We use structure prediction to classify fibers into 276 structural clusters and 1177 domain fold families, which likely mediate glycan and protein binding on the cell surface of eukaryotes or bacterial targets. DNA sequences encoding these rapidly evolving domains may have been acquired from diverse eukaryotes, bacteria, and viruses. Finally, we experimentally show that a candidate tail fiber from a *Paenibacillus* eCIS can bind and direct effector injection into THP-1 human monocyte-like cells, possibly binding D-mannose on the cell surface. This study reveals the exceptional diversity of eCIS receptor binding domains, suggests new eCIS target cells, and provides thousands of proteins that can adhere to different cell types.

Extracellular contractile injection systems (eCISs) are a class of bacteriophage tail-like protein delivery systems encoded in bacteria and archaea that affect virulence and host development, and potentially intermicrobial interactions[1–5]. eCISs are utilized by bacteria for delivery of various cargo (effector) proteins, mostly toxins, to recipient cells and are emerging as a promising biotechnological tool for modular protein delivery[6]. eCIS cargo proteins display highly adaptive versatility with an N-terminal signal-dependent effector loading mechanism[7]. Initially discovered as insecticidal agents in *Serratia entomophila*, eCIS

were denoted as antifeeding prophage (Afp), that causes the Amber disease in the New Zealand grass grub[3,8]. Similar eCIS operons were independently identified in entomopathogenic nematode symbionts as the *Photorhabdus* virulence cassettes (PVC), which delivers insecticidal toxins[1,9], and in *Pseudoalteromonas luteoviolacea*, which induce metamorphosis of its host, the marine tubeworm[2,10]. eCIS complexes have been recently identified in *Algoriphagus machipongonensis*[5], which contain unique structural features. Notably, some eCIS related systems function intracellularly, such as in *Streptomyces*[11,12], where they

[1]Department of Plant Pathology and Microbiology, Institute of Environmental Science, The Faculty of Agriculture, Food, and Environment, The Hebrew University of Jerusalem, Rehovot, Israel. [2]NHC Key Laboratory of Systems Biology of Pathogens, Beijing, China. [3]Key Laboratory of Pathogen Infection Prevention and Control (MOE), Beijing, China. [4]State Key Laboratory of Respiratory Health and Multimorbidity, Beijing, China. [5]National Institute of Pathogen Biology, Chinese Academy of Medical Sciences & Peking Union Medical College, Beijing, China. [6]These authors contributed equally: Nimrod Nachmias, Zhiren Wang. ✉e-mail: jiangfenguva@126.com; alevy@mail.huji.ac.il

deliver toxins for self-elimination in response to stress, and in the multicellular cyanobacterium *Anabaena*, which is thylakoid-anchored (denoted as thylakoidCIS, tCIS)[13] via membrane bound tail-fibers. These fascinating discoveries highlight CIS's versatility and significance within various microbes, both extracellularly and intracellularly.

eCIS-related genomic loci have been extensively studied computationally, which involved analysis of taxonomic and ecological distribution and classification of various eCIS[4,14]. A wide genomic analysis revealed that approximately 2.2% of sequenced bacteria and archaea have been found to contain these systems, with a high prevalence in terrestrial, aquatic, and invertebrate-associated bacteria and unexplained depletion in mammalian pathogens[4]. Nevertheless, identification of tail fiber genes, which are encoded by genes such as Afp13 and Pvc13 of Afp and PVC, respectively, is lacking in most eCIS operons. This missing piece is significant, being the putative target specificity determinant of these systems, likely adhering to specific receptors. Identifying tail fiber genes is challenging due to their rapid diversification[15] under selective pressure to recognize varied targets[16]. This sequence divergence, coupled with their inherent flexibility preventing structural resolution, complicates traditional homology-based discovery.

Here, we addressed this challenge by utilizing a domain-based computational approach to identify rapidly evolving tail fiber genes, discovering 3445 tail fibers across 2585 eCIS systems from 1069 microbial genomes. Our analysis revealed five novel N-terminal baseplate anchor domains and categorized the fibers into 267 structural clusters comprising over 1000 domain fold families. Functional validation showed that receptor binding domain from *Paenibacillus sp. URHA0014* tail fiber that was engineered into PVC complex enabled specific targeting of the human monocyte-like cell line THP-1. Pretreatment with D-mannose or mutations in key binding residues in the engineered tail fiber inhibit the target cell recognition of engineered eCIS, pointing to a plausible mechanism of surface glycan binding. These findings illuminate eCIS target specificity mechanisms across the bacterial tree of life, and establish a foundation for engineering programmable protein delivery systems.

## Results

### Identification of conserved domains leads to discovery of novel eCIS tail fiber genes

Tail fibers likely determine eCIS target specificity and have the potential to elucidate target identity of largely unexplored eCIS particles. This study aimed to maximize the identification of novel eCIS tail fiber gene candidates to allow characterization of target cell binding machinery. First, we searched the eCIStem database, comprising 1425 eCIS operons across 1249 bacterial and archaeal genomes[4]. Our initial sequence similarity search identified 629 genes resembling Afp13 and Pvc13, covering only 44% of eCIS operons, but their homology was partial and of low-confidence. Structural predictions using AlphaFold2-multimer[17,18] revealed that Afp13 resembles a human adenovirus shaft with a beta-spiraled shaft and Ig-like knob, while PVC13 exhibits a chimeric structure of the same shaft with a phage needle-like C-terminal region (Fig. 1a–c), as was solved in the course of our study[19]. These findings highlight the modular nature of tail fiber genes. To address this inherent variability feature, we used all-against-all BLAST[20] and MMseqs[21] clustering to generate Hidden Markov Models (HMMs)[22] of conserved protein domains (Fig. 2a and Supplementary Fig. 1). This led to the identification of novel tail fiber domains. The domain organization suggested high conservation of N-terminal segments, thus we used these regions in *jackhmmer* search[22] and retrieved 1269 additional fiber genes, nearly doubling the number of loci harbouring fibers.

Analysis of retrieved genes revealed five new conserved N-terminal domains likely serving as baseplate attachment sites, that we termed: eCIS Baseplate Anchor Protein (eBAP1-5) domains (Fig. 2b). Striving to maximize tail fiber gene candidates, we expanded the search to a non-redundant combined database which contains 86,785 non-redundant bacterial and archeal genomes. We identified 2585 putative eCIS genomic loci using a Pfam-based scoring system that prioritized clusters with multiple eCIS-enriched domains (Supplementary Data 1,2), with higher scores given to distinctive Pfams such as DUF4157, Pvc16_N, and CIS_tube. This yielded 3445 fiber-encoding genes while excluding loci with T6SS related Pfams (Supplementary Data 1). We then divided the fibers by N-terminal domains into five groups: eBAP1 and eBAP2 (~50 amino acids in length) are structurally related and are present in 362 and 603 gene candidates, respectively (Fig. 2c). A previously unknown ~200-amino-acid domain, found in 1420 genes, contains a shoulders-like[23] structural feature like the tCIS crown domain, and was designated eBAP3. The first 200-amino acids of the tCIS crown gene, widespread in 402 genes, were classified as eBAP4 (Fig. 2d). Additionally, the DUF6519 domain (eBAP5) was identified in 658 genes, sharing partial similarity with eBAP3/eBAP4 but featuring a distinct 200-amino-acid insertion (Fig. 2e). Its presence in 1114 genes in the InterPro database suggests it may serve as a novel tail

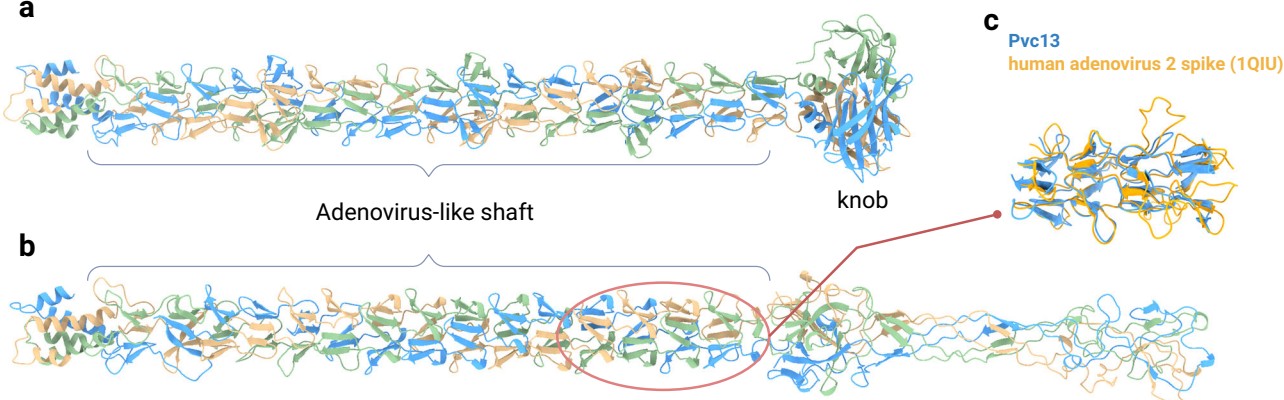

**Fig. 1 | Structural prediction and analysis of Afp13 and Pvc13 proteins.** Structural prediction of AFP13 (**a**) and PVC13 (**b**) in trimeric form. Both genes contain a shaft region characterized by adenovirus-like repeats forming a characteristic beta-spiral domain. Afp13 seems to harbor a Ig-like knob domain reminiscent of adenovirus spike, while in Pvc13, this region seems to be swapped by phage collar domain forming a unique chimeric protein. **c** a Foldseek-multimer result based structural comparison of Afp13 with human adenovirus spike protein (PDB:1QIU). All structures presented in figures are displayed with per-residue pLDDT scores and corresponding PAE plots in Supplementary Fig. 11. Created in BioRender. Levy, A. (2025) https://BioRender.com/q2rg02f.

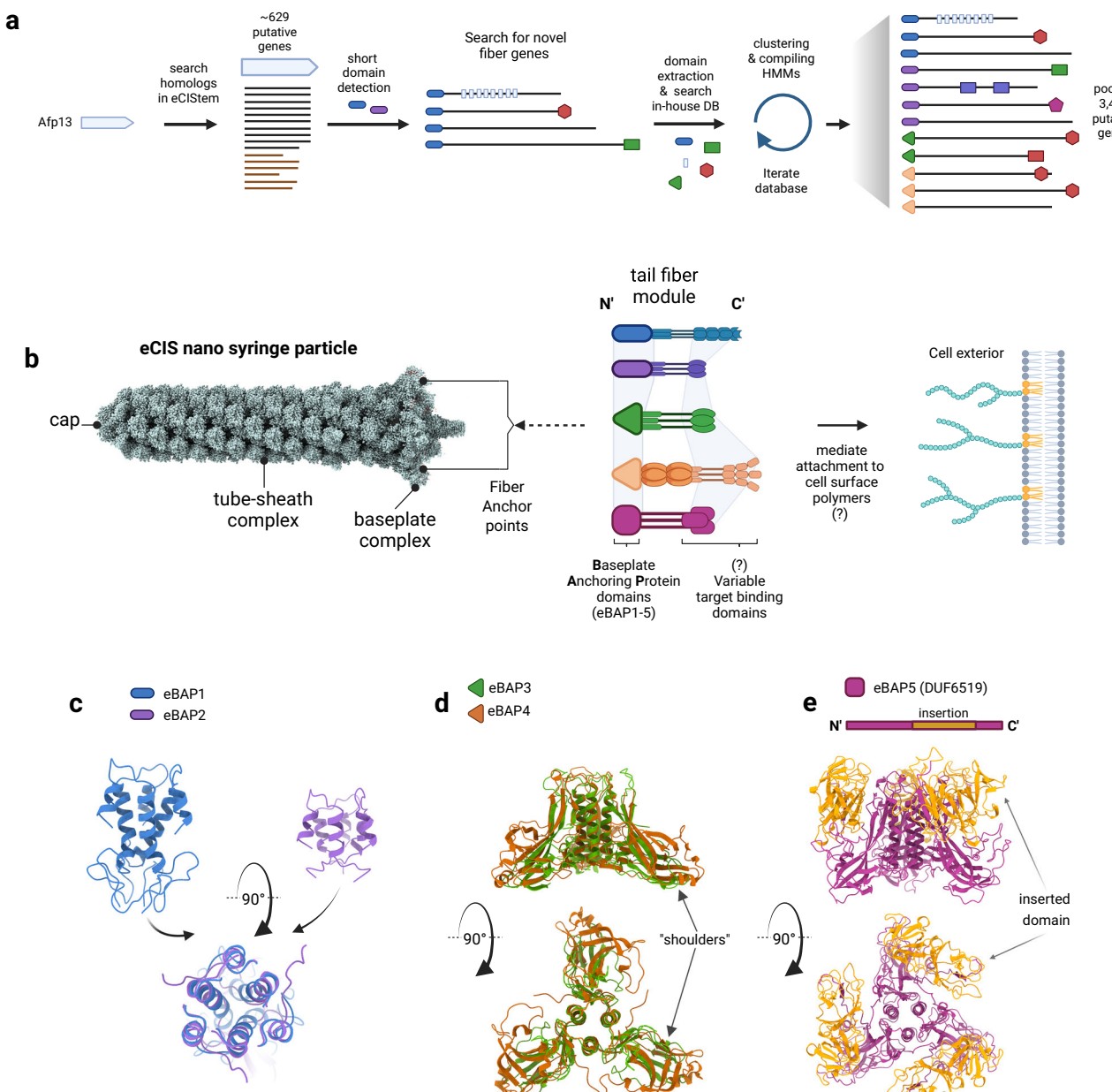

**Fig. 2 | Computational identification and characterization of novel tail fibers and their constituting domains. a** Bioinformatics workflow for identifying novel eCIS tail fiber genes. Known tail fiber sequences were used to search and generate HMM profiles of conserved domains based on the eCIStem database. N-terminal domains were used in *jackhmmer* iterative searches against the nr combined database retrieving 3,2445 novel tail fiber candidates harboring five major N-terminal baseplate anchor domains termed eBAP1-5 (eCIS Baseplate Anchor Proteins) and various domains. **b** Schematic representation of an eCIS particle reconstructed from the PVC solved structures (PDB IDs 6j0b,6j0f,6j0m,6j0n)[51] with tail fibers containing different eBAP domains at their N-termini anchoring them to the baseplate hub. The baseplate acts as a mounting point for the tail fibers to project outwards and engage target receptors on recipient cells via their C-terminal binding modules. **c** Structural prediction and comparison highlighting the conserved bundled α-helical folds of the eBAP1 (from PVC13, blue) and eBAP2 (from IMG gene 2505186636, purple) baseplate anchor protein domains, which share structural similarity despite distinct sequences (structure overlay below). **d** Predicted trimeric structure overlay comparison of a representative eBAP3 domain (from IMG gene 2541268533, green) from eBAP3-containing tail fiber with eBAP4 domain (from tCIS crown gene, PDB:7b5h, brown). Shoulders-like domains are shown with arrows. **e** Domain scheme representation with structure prediction of eBAP5 (DUF6519, magenta) with the distinctive inserted (evolved) domain (highlighted in orange in gene diagram and 3D predicted structure). **a**, **b** were created in BioRender. Levy, A. (2025) https://BioRender.com/q2rg02f.

fiber marker in phage or other systems as well. These findings expand our ability to characterize eCIS evolution and examine microbial host specificity. By defining five novel eBAP domains (Supplementary Data 3–7), we establish a foundation for future phylogenetic, structural, and functional analyses, closing the gap on this elusive structural feature.

## eBAP domains show congruence with ancient evolutionary lineages and specific niche adaptations

We further explored the distribution of tail fibers across eCIS phylogeny in the light of the newly discovered eBAP domains. Our eCIS phylogeny analysis, based on the phylogenetic tree of the conserved structural protein Afp8 amino acid sequence (which encodes a VgrG

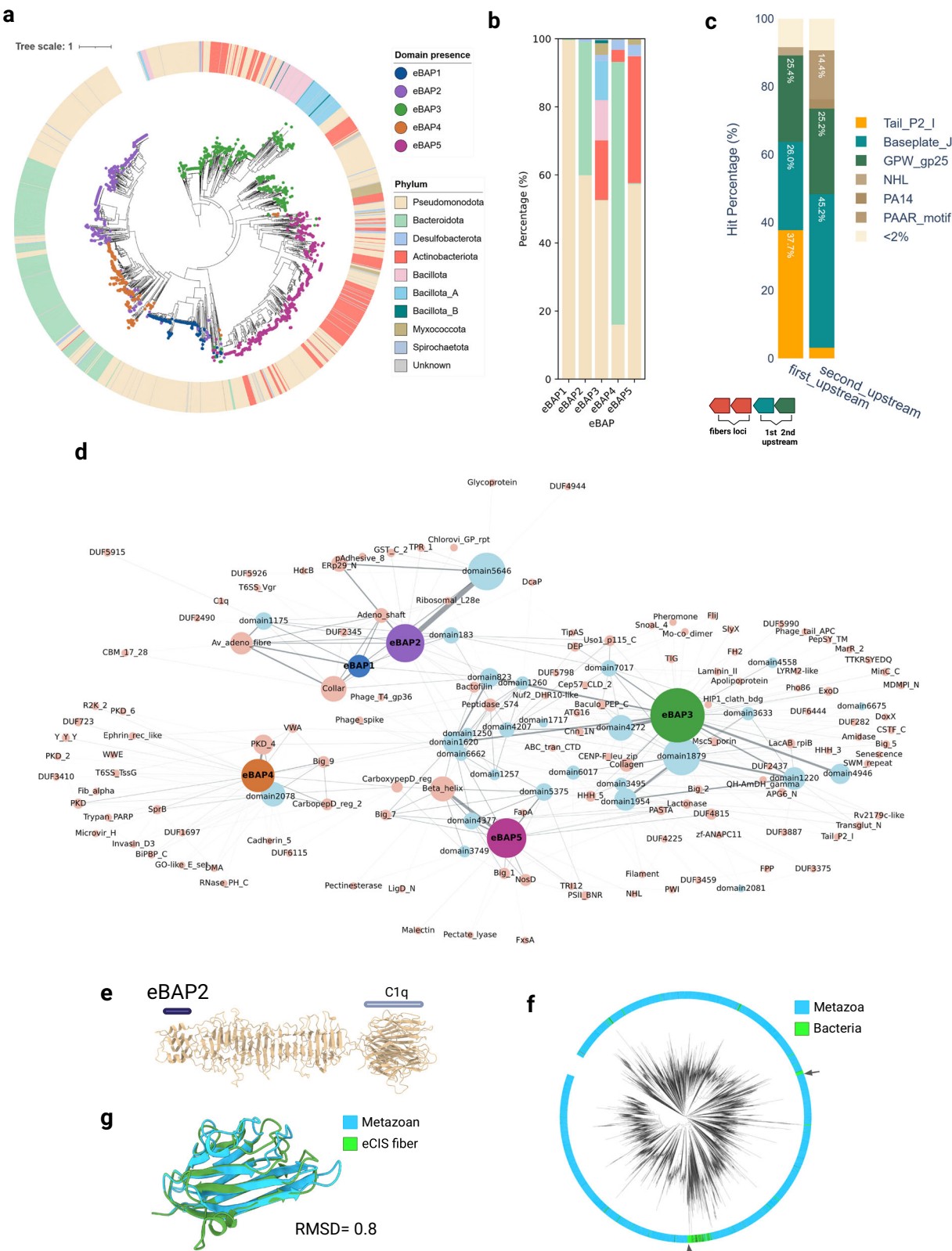

protein), showed eBAPs are congruent with the structural evolution of eCIS (Fig. 3a). The eBAP1-5 containing eCIS loci were grouped and dispersed each to specific branches, indicating five major events of structural and possibly functional divergence in eCIS evolution, that were likely followed by horizontal gene transfers, as shown by the diverse phyla present in each clade (Fig. 3a). We further dissected each eBAP related clade (Fig. 3b). All eBAPs are present in at least some

Pseudomonodota eCIS. eBAP1 is exclusive to Pseudomonodota. eBAP2 and eBAP4 are also distributed in Bacteroidota and slightly in Cyanobacteriota, in addition to Pseudomonodota. eBAP3 is evenly distributed in Bacillota and Actinobacteriota, while eBAP5 is correlated with Actinobacteriota. Notably, finding large amounts of fiber genes in Actinobacteriota systems is intriguing as it was previously assumed, based on *Streptomyces* CIS, that this phylum relies on intracellular CIS

**Fig. 3 | Phylogenetic distribution and evolutionary dynamics of the tail-fiber genes. a** eCIS phylogenetic tree compiled from the *afp8* (VgrG) gene sequences. The circles at the branch tips represent the presence of eBAP1-5 domains, displaying grouped distribution. The outer ring shows bacterial phylum of the bacterial genome harboring the system, displaying horizontal transfer. **b** A stacked bar-plot displaying the normalized share of phylum from (**a**), divided by eBAP groups. Color scheme as in (**a**). **c** A stacked bar-plot displaying the gene functions found upstream to the fiber genes, indicating their conserved genomic neighborhood and putative baseplate adaptors. Baseplate_J (parallel to Afp11), GPW_gp25 (Parallel to Afp12), and Tail_P2_I Pfams stand-out as putative baseplate binding partners (Supplementary Fig. 3). **d** Network representation of eBAP1-5 and Pfam domains found on the fiber genes. Node size represents the number of

hmmscan hits found on the fiber genes, an edge is drawn each time domains are found on the same gene, while edge thickness represents number of connections. eBAPs are colored by their unique colors, 37 discovered sequence-based domains are in light blue and Pfam domains are in pink. **e** C1q domains as an example of a HGT candidate. Structural prediction of a C1q harboring fiber gene, domains are marked with lines above identified domains. **f** Phylogenetic tree made from all domain sequence alignment of the C1q domains in Pfam database (interpro API) showing the bacterial branches containing two types of eCIS fibers are nested within eukaryotic clades. **g** AF2 predicted structure comparison of to C1q domains found adjacent on the tree. From eCIS fiber (IMG gene ID:2606531960) and from a metazoan domain (Uniprot ID A0A9D4MG84) from *Dreissena polymorpha* (a Zebra mussel).

---

alone, which should lack tail fibers[11,24]. Nevertheless, loci that lacked fibers gene candidates might act intracellularly or through mechanisms yet to be discovered, and thus were left out of the scope of this study.

We further investigated the fibers occurrences and genomic neighborhoods to test their position conservation within the operon by evaluating the fiber loci within eCIS operons and their upstream genes. Out of 2585 gene loci that contain fibers; 23.4% contain more than one fiber gene per loci with the maximum being five fibers per loci. Phyla enrichment analysis shows *Myxococcota* and *Pseudomonodota* are enriched in multi-fiber eCIS loci, while *Bacilota* and *Desulfobaterota* are depleted, suggesting this trait is associated with niche adaptation (Supplementary Fig. 2a–c). We examined sequence and predicted structural similarity in multi-fiber loci. We found instances of both structural resemblance yet with remarkable sequence divergence and completely diverse sets in terms of domain composition (Supplementary Fig. 2c–e), demonstrating the versatile evolution of eCIS baseplate composition. Further analysis of each specific fiber found in eCIS loci may shed light on its putative role in target binding. The fact that some eCIS operons acquired multiple tail fiber genes or possess evolved duplicates highlights the versatility and rapid evolution of this gene group.

### Phylogenomics and structural analyses reveal two main baseplate types driving distinct eCIS evolutionary trajectories

Operonic organization analysis revealed that fiber genes are positioned downstream of tail_P2_I and Afp11/12 (Baseplate J and gp25) homologs in ~90% of cases (Fig. 3c), consistent with PVC structures showing fiber anchoring to Afp/Pvc12. Using AlphaFold2/3-multimer modeling, we discovered that while most eBAPs anchor to Afp12 homologs, eBAP2 might also utilize binding of Pvc11, and that the Tail_P2_I harboring gene might bind BAP3/eBAP5, resembling R-pyocin fiber attachment mechanisms (Supplementary Fig. 3).

Phylogenetic analysis of Afp11 sequences revealed clustering patterns congruent with eBAP domain distributions while additionally identifying two distinct baseplate architectures (Supplementary Fig. 4a). eBAP types 1, 2, and 4 incorporate both Afp11/12 homologs while lacking Tail_P2_I genes, whereas eBAP types 3 and 5 rely exclusively on Tail_P2_I adaptors and largely lack Afp12 homologs. This clear compositional grouping strongly suggests two major evolutionary trajectories underlying eCIS diversification.

Structural modeling across eBAP groups demonstrated diverse assembly architectures: eBAP1-2 fibers exhibited the backward tilt observed in Afp/PVC systems, eBAP4 anchored at the baseplate bottom, similar to tCIS (Supplementary Fig. 4b–d). eBAP3,5 Tail_P2_I-containing loci seem to utilize loop-mediated attachment accommodate up to four fibers per baseplate, representing an advanced targeting mode. Notably, eBAP3 systems showed striking homology to T6SS baseplate architecture, with Tail_P2_I genes resembling TssG baseplate components and eBAP3 fibers sharing structural similarity with TssK proteins, suggesting shared evolutionary origin

(Supplementary Fig. 4e, f). These findings establish a model of dual baseplate types as key drivers of eCIS structural diversification (Supplementary Fig. 4h).

### Pfam domain analysis reveals functional diversity and possible evolutionary origins of fiber domains

To characterize the functional architecture of fiber genes, we performed HMMscan[25] against the Pfam database[26] and visualized domain co-occurrence within the protein as a network (Fig. 3d). This analysis revealed distinct domain association patterns across the five eBAP groups. Although eCIS operons are characterized by a group of only 11 core domains[4], we identified a striking number of over 200 tail fiber domains: including 165 Pfam domains and 37 novel domains, which we defined based on all against all Blast, that are found within the eCIS tail fiber genes that might be responsible for its versatile shape, size, and receptor binding. eBAP1 domains showed restricted association, primarily linked to Av_adeno_fiber domains found in the well-studied *Photorhabdus* PVC and *Serratia* Afp systems. In contrast, eBAP2-5 domains anchor diverse functional modules. eBAP2 also co-occurred with adenovirus shaft repeat, gp36 from phage T4, C1q domain that is prevalent in animal proteins[27], and the Peptidase_S74 chaperone. The eBAP3 group displayed the most diverse associations with several types of domains. Some domains serve as enzymes, e.g., Peptidase_S74, amidase, and the lactonase domains. Others might imply target binding due to their role in attachment, such as H-lectin, SWM_repeat[28], Laminin_II, BIG_2/5 domains. eBAP4 formed a distinct network module, strongly associating with immunoglobulin-like fold domains including PKD, Big11, von Willebrand factor type A (VWA), and carboxypeptidase-regulatory (PKD) domains as well as Invasin_D3 and Cadherin_II attachment proteins (Supplementary Fig. 5a). Notably, the overall fabric of the network seems to be interconnected by Pfam and newly defined domains, some of which are abundant, suggesting high domain swapping and/or frequent acquisitions.

Next, we detected the phylogenetic origin of the Pfam tail fiber domains. Strikingly, we found 64 Pfam domains that display phylogenetic incongruence with bacteria using a conservative method. Namely, over 90% of proteins in the InterPro taxonomy database are non-bacterial, which might point out potential donors for cross-kingdom acquisitions (Supplementary Fig. 5b). Notably, several bacterial eCIS sequences (e.g., adenovirus shafts, C1q domains) formed nested clades within metazoan and viral phylogenetic branches rather than clustering as sister groups, a pattern consistent with cross-kingdom horizontal gene transfer rather than shared ancestry (Fig. 3e–g and Supplementary Fig. 5b–d). Strikingly, the analysis reveals that eCIS tail fibers evolved functional plasticity through horizontal acquisition of receptor-binding domains from eukaryotes (animals, plants, and fungi), viruses, and phages. Incorporation of immune-related domains (C1q, immunoglobulins, PKD, VWA) suggests bacterial strategies like host immunity mimicking or interference. This modular evolution mirrors phage adaptability, enabling eCIS systems to target diverse hosts through acquired functional modules.

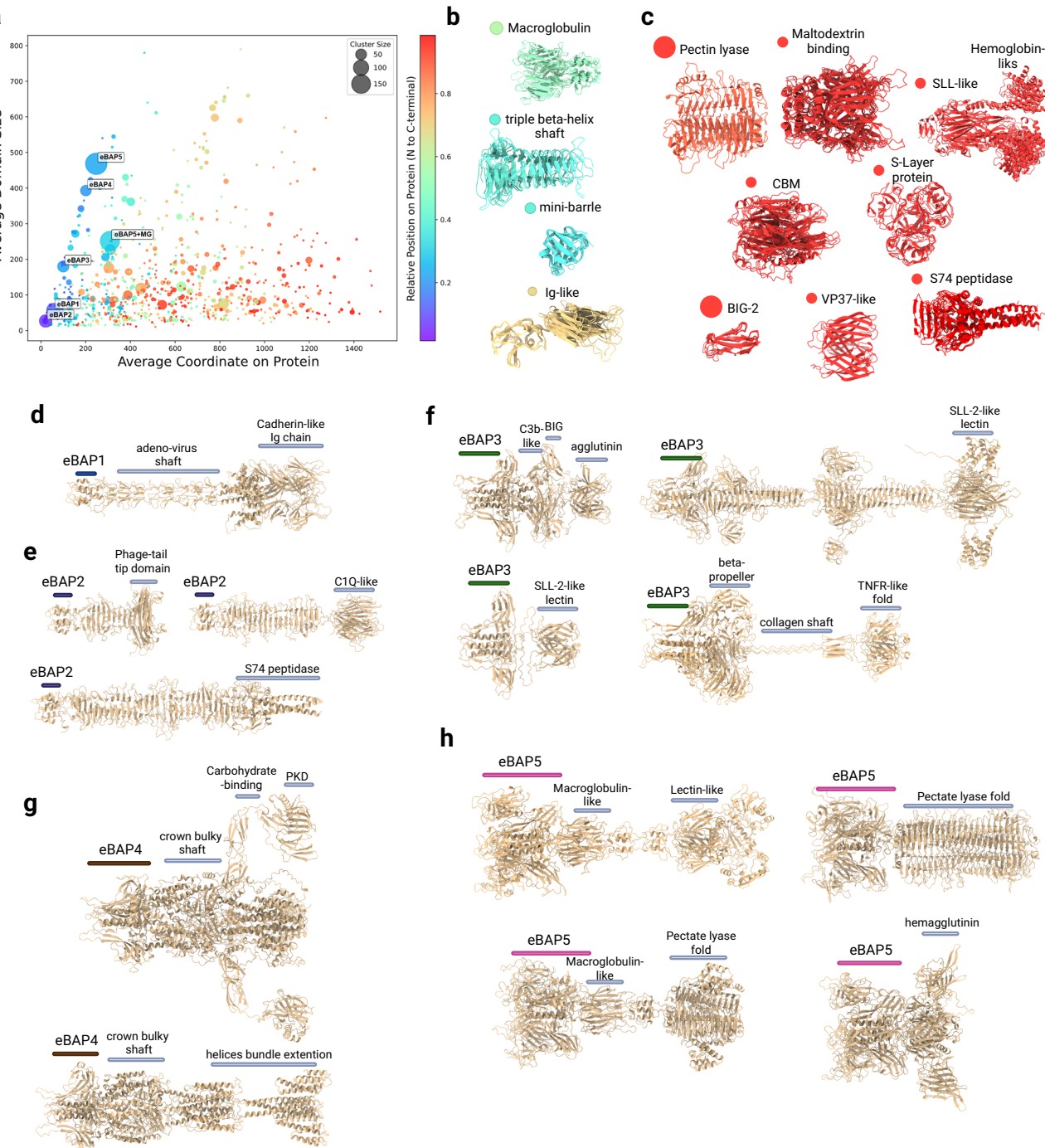

**Fig. 4 | Structural analysis of fiber proteins unveils highly polymorphic modules decorated with various putative carbohydrate and protein binding domains. a** Domain foldseek clusters found on fiber genes in scatter plot representation. Each circle on the plot represents a cluster of domains found on the fibers visualizing cluster size (cycle diameter), relative average positions (rainbow color range from violet to red colors), and midpoint of the domain in obtained coordinates (by average). Labels showing clusters which represent eBAP1-5 N-terminal domains. eBAP5+MG represents instances where eBAP5 gets interwinded with adjacent macroglobulin domains (**b**, **c**), exemplary structure overlays representing the middle region (**b**) and the c-terminal region (**c**). **d**–**h** Examples of predicted structures from prominent structural architectures of eBAP1-5 containing genes. **d** eBAP1: Adenovirus shaft from *Mycetohabitans rhizoxinica* HKI 454 (IMG gene ID 650723730). **e** eBAP2: Mini-fibers from *Derxia gummosa* DSM 723

(2529305320), C1q from *Aquimarina* sp. AU119 (2606531960), S74 from *Microscilla marina* ATCC 23134 (2639240252). **f** eBAP3: C3b/BIG from isolate *An92 spO02159175* (ID: NFGZ01000037_8), SLL lectins from *Pseudoxanthomonas broegbernensis* (2861529714) and *Embleya scabrispora* (KB889561.1_365), beta-propeller/ collagen from *Aromatoleum buckelii* (WTVH01000027.1_19). **g** eBAP4: Bulky shafts from *Cellulomonas* sp000688475 and helical extensions from *Malonomonas rubra* (2588100597). **h** eBAP5: Macroglobulin and lectin -like domains from *Desulfococcus multivorans* (ATHJ01000059_23), Mega pectin lyase from *Rhizobium grahamii* (2535518109), Macroglobulin and pectin lyase from *Pseudogulbenkiania ferrooxidans A* (644377952), and with hemagglutinin from *Thioflavicoccus mobilis* (2507114133). All structures presented in figures are displayed with per-residue pLDDT scores and corresponding PAE plots in Supplementary Fig. 11.

## Structural prediction and classification of fibers and fiber-domains

The complexity and versatility of the observed fibers prompted us to apply structure prediction and alignment tools for comprehensive analysis at two levels: whole-fiber clustering and identification of receptor-binding domains. Our workflow began with clustering the complete fiber database at 70% similarity and coverage, effectively reducing the dataset from 3445 to 1098 representative sequences. Since tail fibers are ordered in trimers[29–31] we predicted trimeric structures for these representatives using AF2-multimer, establishing the eCIS-fiber 3D-database for further analysis. Notably, the AF2 predictions yielded uniformly high-confidence models (pLDDT > 70 in over 85% of structures) with clear domain boundaries, and the trimeric assemblies showed tight, biologically plausible interfaces, validating the structural assignments used throughout this study (Supplementary Fig. 6a, b). Additionally, we assessed model globularity and found eBAP1–2 to form elongated fibers, whereas eBAP3–5 adopt more bulkier architectures (Supplementary Fig. 6c, d).

The structural characterization proceeded through parallel pathways (Supplementary Fig. 6e): Foldseek structural clustering of complete fibers (at 25% sequence similarity and 70% coverage) and detailed structure-based extraction of all globular units and subunits, followed by clustering with equivalent parameters. The whole-fiber analysis produced 263 distinct structures, while our domain-level analysis identified 3515 individual units that further grouped into 1177 fold families decorating the fibers (Fig. 4a and Supplementary Fig. 6f). We then conducted a foldseek search of the novel domains against PDB, which identified a variety of shaft-region fiber domains (Fig. 4b) and putative C terminal receptor-binding domains (Fig. 4c).

We observed that eBAP1, the group that includes tail fibers from the AFP and PVC, almost uniformly carries adeno-shaft repeat folds (Fig. 4d) followed by collar and needle domains, while in some cases the C-termini displays some variety harboring domains with Ig-like cadherin fold not detected by Pfam search. The eBAP2 group features tail fibers with intertwined triple beta-sheeted shafts resembling phage tail fibers (Fig. 4e). Surprisingly, this group encompasses short mini-fibers, which is the most abundant group in our database, harboring putative saccharide-binding domains similar to a receptor binding domain from *Acinetobacter* phage (PDB:6E1R). As described previously, we observed in this group fiber carrying a C1q domain typically associated with human complement systems[27,32] and the S74 fold (a chaperone of endosialidase).

## eBAP3-5 share common shoulders-like substructure yet define distinct groups and putative binding module arsenals

Despite their similar shoulder-like regions resembling phage receptor-binding proteins[23], eBAP3-5 domains differ in sequence identity and overall architecture. We noticed the shoulder-like region backbone chain might be intertwined with adjacent domains (Supplementary Fig. 7) observed in the various subtypes, which might enhance protein stability while promoting evolutionary divergence.

eBAP3 fibers display remarkable structural diversity, from minimal attachment-only modules to elongated phage-like β-sheeted shafts (Fig. 4f). Their diverse binding domains—including SLL2-like lectins, bacterial Ig folds, and hemagglutinin structures—suggest specialized glycan-binding functions. Notably, many eBAP3 fibers combine beta-propeller lectins with collagen-like triple helix shafts connected to TNFR-like domain, likely providing enhanced flexibility during target engagement.

eBAP4 represents the most structurally distinct group, characterized by unique bulky shaft regions first described in tCIS crown proteins[13] (Fig. 4g). These domains typically feature 1–2 large α + β globular domains or α-helical extensions, connected to flexible chains decorated with multiple Ig, lectin, and PKD attachment modules in various combinations and copy numbers. The utilization of flexible adhesion chains represents a unique attachment strategy that poses a challenge to solve using cryo-EM techniques.

eBAP5 domains contain signature insertion forming miniature β-barrel and β-sandwich folds, with shaft regions often composed of macroglobulin-like (MG) domain (single or repeated) (Fig. 4h). In many cases, we observed that the MG shaft region gets intertwined with the eBAP5 domains, which forms a unique structural group (Fig. 4a). Their C-terminal regions frequently display beta-helix pectin-lyase folds alongside specialized lectin and hemagglutinin domains. Collectively, our structural analysis reveals that eCIS particles employ a diverse array of structural framework which includes carbohydrate and protein-binding domains to recognize and adhere to target cell surfaces, providing insights that could facilitate future eCIS engineering applications.

## Engineered fiber-PVC constructs for targeted cell recognition

We hypothesized that the novel tail fiber genes identified in this study could enable engineering of eCIS to achieve programmable cell targeting. Using the PVC particles expressed on a vector in *Escherichia coli* as a modular chassis, we computationally screened and identified three candidate tail fiber proteins from major clusters (Supplementary Data 8). Comparative analysis of these candidates against wild-type fibers revealed conserved baseplate-attachment domains and variable receptor-binding regions. We prioritized fiber protein-Pb from *Paenibacillus sp*. URHA0014 for experimental validation, which represents a widespread structural group hosting eBAP3 and intriguing domains with C3b and hemagglutinin-like folds. Hemagglutinin folds are common glycoproteins found on the surface of viruses that infect human cells, such as influenza[33] and measles[34] and thus we predicted that the *Paenibacillus* tail fiber protein may similarly adhere to human cells. Two additional fibers, fiber-Mr and fiber-Am, were selected as secondary candidates which represent predominant clusters.

To test targeting specificity, we engineered a chimeric fiberPb-PVC system by swapping the predicted receptor-binding protein (RBP) fragment of fiberPb with the cognate fragment of the PVC tail fiber (Fig. 5a). Positive control PVCs harboring the adenovirus 5 knob domain (R7PVC) were loaded with the plant-derived toxin TcsT that targets eukaryotic cells[7]. CCK-8 cell viability assays on THP-1 human cells demonstrated that treatment with fiberPb-PVC-TcsT reduced cell viability by approximately 60% after 48 h vs. empty R7PVC controls ($p = 5.9E-09$; Fig. 5b), with slightly reduced cell killing than the R7PVC-TcsT positive control. Importantly, fiberPb-PVC exhibited no detectable killing of human A549, HEK293T, or HeLa cells (Supplementary Fig. 8a), demonstrating high target cell binding specificity towards THP-1 cells by the engineered PVC. This result is in contrast to R7PVC, which showed low specificity and injected the TcsT toxin into multiple cell lines (Supplementary Fig. 8a).

Western Blotting confirmed proper TcsT loading in both PVC particles (Fig. 5c), while negative-stain electron microscopy observations validated intact assembly of the chimeric PVCs (Fig. 5d). Moreover, we were able to visualize the direct binding of fiberPb-PVC to THP-1 membranes (Fig. 5e, red arrows). We run similar tests with our secondary candidates (Supplementary Fig. 8b–h), but they demonstrated mild toxin injection into the four tested cell lines, re-affirming the specificity of fiberPb towards THP-1 cell line.

## Mannose-dependent inhibition of fiberPb-PVC cytotoxicity

To explore the mechanistic basis of THP-1 cell recognition by the agglutinin-like domain of fiberPb, we examined potential interactions with cell surface carbohydrates. Competitive inhibition assays using putative ligand molecules revealed that pre-incubation of fiberPb-PVC-TcsT with D-mannose increased THP-1 viability by 32% and 44% in a

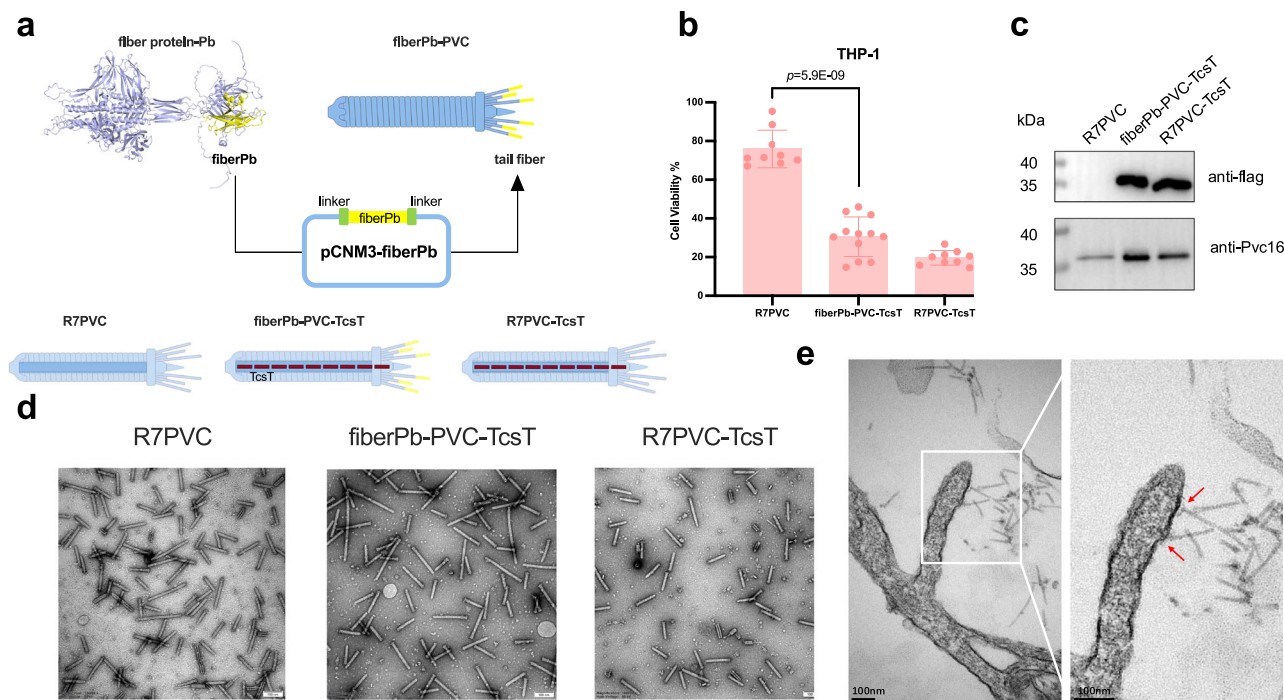

**Fig. 5 | PVC-compatible tail fiber engineering. a** A diagram of the construction of fiberPb-modified PVC. The yellow part represents the predicted receptor recognition fragment from *Paenibacillus sp.* URHA0014 fiber protein. R7 represents the adenovirus 5 (Ad5) binding domain (Ad5-knob(RGD/PK7)). **b** Killing of THP-1 cells by PVC complexes at 0.05 mg/mL concentration after 48 h. Empty R7PVC was used as negative control and R7PVC loaded with TcsT was used as positive control. Graph bars represent mean ± SD, and dots indicate individual data points for $n = 4$ (fiberPb-PVC-TcsT) or 3 (R7PVC, R7PVC-TcsT) biological replicates with 3 technical replicates per experiment. Statistical significance was calculated using two-tailed unpaired corrected Welch's *t*-test. **c** Western blot validation of PVC assembly and protein loading. The presence of the Pvc16 protein indicated that the PVCs were assembled in accordance with the established structure. The flag bands confirmed that the PVCs were loaded with the flag-tagged payload (TcsT). **d** Negative-stain transmission electron microscopy (TEM) graphs of engineered PVC particles. Scale bar, 100 nm. **e** TEM images of THP-1 cells treated with fiberPb-PVC. Scale bar, 100 nm. The red arrows point to engineered fiberPb-PVC bound to the cell membrane. Representative images of 3 independent experiments are shown (**d**, **e**). Source data is provided as a Source Data File. **a** was created in BioRender. Levy, A. (2025) https://BioRender.com/q2rg02f.

concentration-dependent manner ($p = 0.0032$ and $p = 0.0004$, respectively; Fig. 6a), whereas galactose, N-acetylglucosamine, and fucose showed no inhibitory effects (Supplementary Fig. 9a–c). This suggests that D-mannose sugar may interfere with fiberPb-PVC engagement of THP-1 cells.

AlphaFold2 structural predictions and sequence conservation identified three putative target engagement motifs: the VDIT motif (encompassing adjacent sequences VDIT and SGEIVH that form a putative binding pocket), IVF, and K5 (Fig. 6b). Mutating VDIT, IVF, and K5 residues reduced viability to approximately 82%, 57%, and 51%, respectively ($p = 4E-12$, $p = 2.30E-05$, $p = 0.0068$, Fig. 6c and Supplementary Data 10), with VDIT mutation having the most significant effect. Consistent with these experimental results, molecular docking simulations also predicted favorable D-mannose binding at a pocket formed by the VDIT and SGEIVH motifs (Supplementary Fig. 10). Western blots and negative-stain electron microscopy observation confirmed retained TcsT loading (Fig. 6d) and structural integrity across mutants (Fig. 6e). These data imply that the identified motifs may mediate glycan recognition and membrane interactions. While we acknowledge that mannose-based inhibition working via alternative mechanisms (e.g., steric hindrance from mannose binding to THP-1 receptors) cannot be excluded.

Notably, R7PVC-TcsT activity remained unaffected by D-mannose (Supplementary Fig. 9d–f), indicating fiberPb-THP-1-binding specific inhibition. This finding implies a potential structural or functional specificity in the recognition mechanism between fiberPb and D-mannose located on cell surfaces. Further validation via glycan arrays or surface plasmon resonance is required to conclusively establish mannose as a direct ligand for fiberPb.

## Discussion

eCIS have been studied for 20 years now[35] in a small number of prokaryotes with a focus on their caused phenotypes, assembly, 3D structures, and translocated effectors. However, there is hardly any information about the binding mechanisms of eCIS to the target cells[19]. This study reveals the scope of receptor-binding capacity harbored by bacterial eCIS. eCIS achieves target specificity through modular tail fiber gene repertoire with a spectacular structural diversity for a single gene locus.

We defined the evolutionarily conserved novel baseplate-anchoring domains (eBAP1–5) and highly diverse receptor-binding modules. The eBAPs divide the eCIS into five separated eCIS structural subtypes that were massively horizontally transferred within bacterial phyla (Fig. 3a). By combining computational domain discovery with structural prediction, we identified 3445 tail fiber proteins across 2585 eCIS operons, resolving a critical gap in understanding how these systems recognize host cells. Our complex predictions suggest eBAP domains anchor fibers to eCIS baseplates via flexible anchor points, enabling integration of structurally plastic C-terminal architectures most-likely acquired via horizontal gene transfers (HGTs) from viruses, bacteria, and eukaryotes. eBAP3 containing baseplates stood out as putative binding of up to 4 fibers by the unique Tail_P2_I adaptor, suggesting vast utilization of polyvalence by this system subtype.

The clade-specific distribution of eBAP domains reflects ecological specialization: eBAP1-2 exclusivity to versatile Pseudomonodota matches their multi-host and symbiotic lifestyles, while eBAP3's prevalence in soil-dwelling Bacillota and Actinobacteriota challenges assumptions about intracellular-only CIS, suggesting roles in interspecies competition. eBAP2/eBAP4 in Cyanobacteriota highlights the

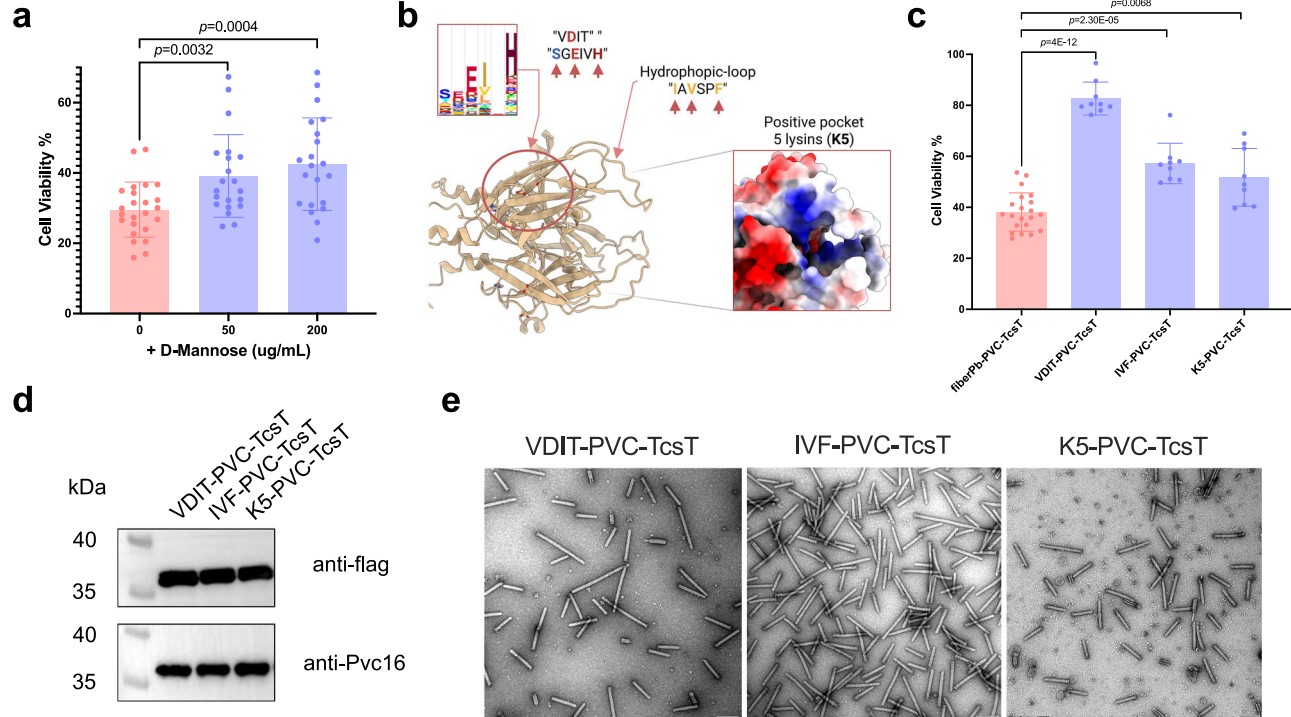

**Fig. 6 | Mannose-based inhibition of target cell recognition. a** Killing of THP-1 cells by 0.05 mg/mL fiberPb-PVC pre-incubated with D-Mannose gradient after 48 h. Graph bars represent mean ± SD, and dots indicate individual data points for *n* = 7 (50 μg/mL, 200 μg/mL) or 8 (0 μg/mL) biological replicates with 3 technical replicates per experiment. Statistical significance was calculated using two-tailed unpaired corrected Welch's *t*-test. **b** Identification of putative functional residues on the fiberPb domain. We identified two conserved motives that form a pocket with VDIT and SGEIVH motifs, a hydrophobic extruding loop with IAVSPF motif and five scattered lysines (Ks) that come together to form a positively charged surface. The bold and colored letters as well as the five lysines mark residues that were mutated for following experiments. **c** Killing of THP-1 cells by fiberPb-PVC and its sugar-binding mutants at 0.05 mg/mL concentration after 48 h. The putative sugar binding sites (VDIT, IVF, K5) were predicted as in (**b**). The two-sided Welch's *t*-test was used for statistical analysis. Sample numbers from left to right: *n* = 21, *n* = 9, *n* = 9, *n* = 9. **d** Western blot validation of TcsT loading of the fiberPb-PVC variants in (**c**). **e** The negative-stain TEM graphs of fiberPb-PVC variants in (**c**). Scale bar, 100 nm. Representative images of 3 independent experiments are shown. Source data is provided as a Source Data File.

---

board utilization of tCIS-like systems that might not strictly anchor to thylakoids, while eBAP5's correlation with Actinobacteriota, which represent a new eCIS subtype previously missed.

Functional validation of *Paenibacillus* fiberPb against a human cell line underscores the biotechnological potential of these findings in examining and redirecting the binding specificity of eCIS. Given that eCIS is encoded by a large variety of environmental bacteria from plants, invertebrates, fungi, soil, and aquatic environments[4], we propose that the endogenous tail fibers can be used for certain environmental and agricultural applications, and perhaps can be used to bind and enable transfer of proteins into eukaryotic cells that lack genetic systems, such as those of different crops, fungi, and insect pests. Competition assays demonstrated that D-mannose treatment increases THP-1 viability in a dose-dependent manner, while mutagenesis of predicted glycan-binding motifs abolished activity. These results align with structural predictions of putative-binding pockets but require validation via direct binding assays. We are looking forward to the elucidation of more binding mechanisms as our large scope identification can be set as a trove for the scientific community.

The structural diversity of eCIS fibers is staggering: clustering revealed 276 distinct fiber types and 1177 domain folds, many resembling eukaryotic immune or adhesion proteins. This suggests eCIS systems employ molecular mimicry to adhere to host cells–a strategy parallel to bacterial and viral cell entry. For instance, eBAP3 fibers combine β-propeller lectins with flexible collagen helices shaft which probably allows flexible movement for connected TNFR-like C-terminal domain, a valuable clue for flexible target engagement yet to be described in the phage or virus literature. eBAP5 fibers incorporate macroglobulin-like repeats, which might play an additional role, such as immune

interference. Such modularity enables niche specialization, as evidenced by eBAP5's dominance in Actinobacteriota, a phylum previously thought to rely solely on intracellular systems. Overall, the eCIS specificity determinant displays plasticity reminiscent of what is observed in organisms involved in evolutionary arms-race such as phage and viruses.

Our eBAP-based approach identified 3445 fiber candidates with estimated 0.23% false positives (8 genes that are clearly non-fiber hits), though it may miss highly divergent variants or incorrectly classify pseudogenes until new anchor types emerge. For researchers who wish to apply our approach to other fields, such as phage tail fibers, we recommend focusing on candidates that: (i) reside in expected genomic positions, (ii) adopt reasonable trimeric conformations computationally, and (iii) display coherent domain architectures, optimally followed by experimental testing. As eCIS fibers undergo rapid diversification, our database requires periodic expansion and filtering to encompass emerging sequence and structural variants.

This work establishes eCIS tail fibers as versatile scaffolds for synthetic biology. We acknowledge the lack of solved atomic structures validating our findings. Future efforts should prioritize resolving atomic-level whole-fiber and fiber-receptor interactions and leveraging the 1177 identified domain folds for therapeutic and biotechnological design. By bridging evolutionary insights with functional validation, we provide a roadmap for harnessing eCIS diversity in targeted protein delivery and ecological studies.

## Methods
### Computational identification of tail fiber domains
To systematically identify conserved domains in eCIS tail fibers, we developed a multi-stage computational workflow. Initial sequence

similarity searches using all-versus-all BLAST (e-value 1e-10, -ungapped, filtered for low coverage of 50% and fragment size between 10 and 500) identified candidate fiber genes across all gene candidates. Conserved regions were subsequently clustered using MMseqs2[21] and CD-HIT[36] at gradient identity thresholds (40%–30%) to account for sequence divergence. Clusters with less than 5 members were discarded. Multiple sequence alignments generated via Clustal Omega formed the basis for constructing HMMs using HMMER[37], which were integrated with the Pfam 35.0 database[38] through hmmpress. The combined HMM database was scanned with hmmscan (e-value ≤ 1e-5), and domain boundaries were resolved using the CATH[39] structural classification toolkit (cath-resolved-hits tool).

To ensure the reliability of this expanded dataset, a multi-layered validation strategy was implemented. First, rigorous quality control was applied during HMM construction to ensure the use of high-quality seed sequences. Second, retrieved candidate sequences were subjected to manual inspection and structural validation. Trimeric structural models were predicted for selected representative domain sequences using AlphaFold2-multimer, structures displaying biologically plausible and consistent folds were chosen. Third, the genomic context of each candidate was analyzed, which confirmed that approximately 90% are located in the expected operonic position downstream of Afp11/12 or Tail_P2_I baseplate homologs. Finally, to estimate the false-negative rate, eCIS loci that did not yield a fiber hit were manually inspected, and no alternative fiber-like genes were found in the expected genomic neighborhood, suggesting these are likely fiberless systems. This comprehensive, multi-layered approach provides robust evidence for the reliability of the identified fiber candidates.

### Identification of eCIS clusters within nr combined database
For comprehensive eCIS identification in our non-redundant combined database, we employed a Pfam-anchored search strategy. We screen genomes for gene clusters (max. 5 genes apart) housing Pfams found to be enriched in eCIS operons in a previous study[4]. We scored putative eCIS loci by Pfam presence and gave a higher score for marker eCIS Pfams such as DUF4157, Pvc16_N, CIS_tube, DUF6519, etc (see Supplementary Data 1). We then pooled all amino acid sequences of genes in loci range plus five genes from the two edges. We searched the five newly defined baseplate-anchoring (eBAP1-5) domains as markers using *jackhmmer* iterative searches (set to 3 iterations), enabling detection of rapidly evolving fiber genes that traditional homology searches often miss. This approach successfully identified 2585 eCIS clusters containing 3445 fiber-encoding genes across 1069 microbial genomes. Clusters lacking identifiable fiber genes or containing capsid (0) or T6SS (18) Pfams were manually excluded as potentially non eCIS (usually found near a true loci).

### Phylogenetic analysis
To examine evolutionary relationships of eCIS tail fibers, we constructed a phylogenetic tree using Afp8 (VgrG protein) sequences, a highly conserved component showing highest presence across nearly all identified eCIS loci and previously proven to reliably represent eCIS phylogeny, similarly to the afp11 core gene[4]. Sequences were aligned with Clustal-omega[40], and a maximum-likelihood tree was built with IQ-TREE[41] using the LG + I + G4 substitution model. Branch support was assessed through 1000 ultrafast bootstrap replicates. The resulting tree revealed clade-specific distribution patterns of the five eBAP domains, using ITOL[42] website with tree display datasets derived from *jackhammer*[22] results, followed by cath-reloved-hits filtering, suggesting distinct evolutionary trajectories. Phylogenetic lineage of genomes were retrieved from our database enabling domain-based phylogenetic profiling to identify potential horizontal gene transfer events between taxonomically distant microorganisms.

### Examination of fiber genes evolutionary dynamics
Examination of multi-fiber operons (23.4% of loci) through pairwise alignment and structural comparison revealed both gene duplication events with subsequent diversification and acquisition of entirely different fiber architectures within the same operon, highlighting the dynamic evolutionary processes shaping eCIS target specificity.

Genomic neighborhood analysis of genes upstream to fiber loci was conducted by evaluating the dominant strand that encodes most of the eCIS loci (most eCIS operons display operonic directionality) pooling both upstream genes and screening them against the Pfam database using hmmscan[43].

### Domain co-occurrence network construction
To analyze functional relationships between Pfam domains in eCIS tail fibers, we constructed a domain co-occurrence network using Python's NetworkX library (version 2.8.4)[44]. First, we performed hmmscan (HMMER 3.3.2) against the Pfam database (version 35.0)[38] with fiber protein sequences using an e-value threshold of 1e-5 and bitscore of 12. Domain hits were filtered to remove overlapping regions, prioritizing matches with higher bit scores when domains competed for the same region. For network construction, each Pfam domain and newly identified eBAP domain was represented as a node. An edge was drawn between two domains if they co-occurred within the same protein sequence. Edge weights were calculated based on the frequency of co-occurrence. Node sizes were scaled proportionally to the number of occurrences of each domain across all fiber genes, and edge thicknesses reflected co-occurrence frequencies. Node colors were assigned based on domain classification: eBAP domains (five distinct colors), novel sequence-based domains (light blue), and Pfam domains (light red). The network was visualized using a force-directed layout algorithm (Kamada-Kawai[45]) to emphasize meaningful interactions of functional domain clusters.

### Identification of potential horizontal gene transfer events
To identify Pfam domains potentially acquired through horizontal gene transfer (HGT), we evaluated the taxonomic distribution of protein family members for each domain. We first extracted all protein sequences and taxonomic lineages associated with each Pfam domain from the Pfam database and retrieved their taxonomic classifications from the InterPro[46] host API.

Domains were flagged as potential HGT candidates when their taxonomic distribution showed phylogenetic incongruence—defined as cases where >90% of protein family members belonged to a non-bacterial kingdom despite being found in our bacterial fiber dataset. For each candidate domain: We calculated kingdom-level representation ratios by dividing the number of sequences from each kingdom (Bacteria, Archaea, Eukaryota, Viruses) by the total number of sequences associated with the domain. We performed sequence alignment between our fiber domain instances and representative sequences from the dominant kingdom using Clustal-omega[40]. We constructed maximum-likelihood phylogenetic trees using IQ-TREE[41] with LG + I + G4 model selection to visualize the phylogenetic relationships between bacterial and non-bacterial sequences. The generated tree was presented with a taxonomic dataset using the ITOL website[42].

### Phylogenomics analysis and complex structure predictions
To elucidate the evolutionary trajectories of eCIS baseplate architectures, a combined phylogenomic and structural modeling approach was used. First, the operonic organization of all 2585 identified eCIS loci was analyzed to determine the genomic position of fiber genes relative to homologs of key baseplate components, including Afp11, Afp12, and Tail_P2_I. A phylogenetic analysis of the baseplate protein Afp11 was then conducted by aligning sequences with Clustal Omega

and constructing a maximum-likelihood tree in IQ-TREE, with branch support calculated from 1000 ultrafast bootstrap replicates. This tree was annotated with the corresponding eBAP type and the presence or absence of the Tail_P2_I gene for each locus to map the distribution of different baseplate compositions. To investigate the physical underpinnings of these architectures, AlphaFold-Multimer was used to model the protein-protein interactions between representative eBAP domains and their putative baseplate adaptors. Finally, structural models of entire baseplate-fiber assemblies were generated for each major eBAP group to assess their overall conformation, and Foldseek was used to perform structural homology searches against the PDB to identify evolutionary links to other systems, such as the T6SS.

## Structural prediction and domain detection

To elucidate the three-dimensional architecture of eCIS tail fibers, we employed AlphaFold2-multimer[17] to predict trimeric structures for 1098 representative sequences (clustered at 70% similarity from the full dataset of 3445 proteins). Trimeric modeling reflected the natural oligomeric state of viral fiber proteins and significantly improved prediction quality compared to monomeric predictions and is widely practiced[29–31]. For comprehensive domain characterization, we implemented dual structural analysis workflows: whole-fiber clustering using Foldseek (25% sequence similarity threshold) yielded 276 distinct structural groups, while extraction and clustering of individual domains identified 1177 fold families.

For protein domain dissection, we designed an algorithm which identifies domains by analyzing secondary structure interactions through a graph-based approach. First, it extracts helices and β-strands using DSSP[47] and represents them as nodes in a graph. Each residue on the secondary structures is assessed for its closest interacting residue (best friend), edges therefore representing spatial proximity of selected interacting residues (8 Å threshold between Cα atoms). Connected components in this graph represent a cluster of secondary structure interacting in the manner described and therefore define potential domains, with further refinement through overlap analysis and size filtering. Prediction confidence was assigned using pLDDT scores outputted in AlphaFold models[48]. The tool outputs domain boundaries in tabular format, enabling systematic characterization of predicted structures and facilitating downstream functional analysis. This approach effectively identifies structural domains without relying on sequence homology, making it particularly valuable for novel protein families. To extract (or dissect) protein domains from AlphaFold-predicted structures, we developed a Python-based extraction pipeline utilizing BioPython's PDB module[49]. The extraction pipeline filters redundant domains with similar boundaries (±4 residue positions), prioritizing regions with higher confidence scores. Each domain is saved as an individual PDB file with preserved residue numbering and chain identifiers, enabling direct mapping to original coordinates. These predicted structures were clustered and compared against the Protein Data Bank[50] using Foldseek's threading-based algorithm to identify structural homologs and infer potential receptor-binding functions. This approach overcame limitations of sequence-based methods for these rapidly evolving proteins, enabling identification of domains with structural similarity to carbohydrate-binding modules and receptor-binding proteins despite low sequence conservation. AlphaFold metrics of the different proteins are provided in Supplementary Data 12.

## Bacterial strains and cell culture

*Escherichia coli* strains DH5α and EPI300 were cultured in Luria-Bertani (LB) broth at 37 °C with shaking at 200 rpm. Mammalian cell lines were maintained at 37 °C in a humidified 5% $CO_2$ atmosphere. HeLa and HEK293T cells were cultured in Dulbecco's Modified Eagle's Medium (Gibco) supplemented with 10% fetal bovine serum (FBS, Gibco), while A549 and THP-1 cells were cultured in RPMI 1640 medium (Gibco) supplemented with 10% FBS.

## Plasmid construction

Predicted tail fiber fragments were commercially synthesized (Tsingke Biotechnology, China) and cloned into the PVC-expressing plasmid pCNM3 (see Supplementary Data 9). Site-directed mutations were introduced using homologous recombination. PCR amplification was performed using Tks Gflex™ DNA Polymerase (Takara), and fragments were assembled using T4 DNA Ligase (New England Biolabs). To produce functional PVC complexes, EPI300 cells were co-transformed with three plasmids: modified pCNM3_fibers (structural components), pBR-LysR (regulatory elements), and pBBRN_TcsT (TcsT toxin payload). Control strains were generated with either empty pBBRN (negative control) or wild-type pCNM3 with pBBRN_TcsT (positive control). Plasmids, primers, and sequences are listed in Supplementary Data 9 and 10, respectively.

## PVC purification

PVC complexes were purified according to previously described methods with modifications. Briefly, transformed EPI300 cells were grown in 200 mL LB broth at 30 °C with shaking at 200 rpm for 24 h. Cells were harvested and lysed in 12 mL buffer P (25 mM Tris pH 7.4, 140 mM NaCl, 3 mM KCl, 200 µg/mL lysozyme, 50 µg/mL DNase I, 0.5% Triton X-100, 5 mM $MgCl_2$, and 1× protease inhibitor) for 45 min at 37 °C. After centrifugation at 20,000 × g for 6 min, the supernatant was ultracentrifuged at 200,000 × g for 75 min at 4 °C. The pellet was resuspended in 1 mL ice-cold PBS and centrifuged at 20,000 × g for 10 min at 4 °C. The resulting supernatant underwent a second ultracentrifugation at 200,000 × g for 75 min at 4 °C. The final pellet was resuspended in 400 µL ice-cold PBS, clarified by centrifugation at 20,000 × g for 10 min at 4 °C, and the supernatant containing PVC particles was stored at 4 °C.

## PVC verification

PVC loading was verified by Western blotting. Purified PVC complexes were mixed with 2× Laemmli Sample Buffer (Bio-Rad), heated at 99 °C for 10 min, and separated on 12% SDS-PAGE gels (Vazyme) at 150 V for 1 h. Proteins were transferred to PVDF membranes, probed with appropriate antibodies (Supplementary Data 2), and visualized using a Bio-Rad ChemiDoc system.

## Cell viability assays

Cell viability was assessed using the Cell Counting Kit-8 (CCK8, MedChemExpress). HeLa, HEK293T, and A549 cells were seeded in 96-well plates at $1 \times 10^5$ cells/mL and incubated overnight. THP-1 cells were seeded at $1 \times 10^6$ cells/mL and differentiated with 0.1 µg/mL phorbol-12-myristate-13-acetate. Cells were treated with purified PVC complexes (50 µg/mL) for 48 h at 37 °C. Culture medium was then replaced with fresh medium containing CCK8 solution and incubated for 1 h at 37 °C. Absorbance at 450 nm was measured using a TECAN microplate reader.

## Electron microscopy

**PVC assembly verification.** PVC assembly was confirmed by negative staining electron microscopy. Briefly, 5 µL of purified PVC sample was applied to a glow-discharged holey-carbon-coated copper TEM grid (Quantifoil) for 60 s. After removing excess liquid, the grid was stained twice with 2% uranyl acetate and air-dried at room temperature before imaging with an FEI Tecnai (G2 Spirit TWIN) electron microscope at 80 kV.

## Cell-binding visualization

To visualize PVC binding to target cells, cells were seeded in 12-well plates and incubated with PVC complexes (0.5 mg/mL) for 3 h. After gentle scraping, the cells were collected and centrifugation at 100 × g for 5 min, the supernatant was replaced with 2.5% glutaraldehyde for fixation. The pellet was fixed for 2 h at room temperature and stored at

4 °C. Following dehydration, resin infiltration, embedding, and ultra-thin sectioning, samples were placed on glow-discharged carbon-coated gold TEM grids, stained with 2% uranyl acetate, and imaged using an FEI Tecnai (G2 Spirit TWIN) electron microscope at 100 kV.

## Reporting summary

Further information on research design is available in the Nature Portfolio Reporting Summary linked to this article.

## Data availability

Raw data: https://zenodo.org/records/15274396 https://zenodo.org/records/17150571 eCIS fibers 3D database and clustered domains: https://zenodo.org/records/15277912. PDB accessions mentioned in the paper: https://www.rcsb.org/structure/1QIU. https://www.rcsb.org/structure/6J0B. https://www.rcsb.org/structure/6J0F https://www.rcsb.org/structure/6J0M https://www.rcsb.org/structure/6J0N https://www.rcsb.org/structure/7B5H https://www.rcsb.org/structure/6E1R. Source data are provided with this paper.

## Code availability

https://github.com/Nimrod198/eCIS_tail_fibers.

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

## Acknowledgements

A.L. is generously supported by the Israeli Ministry of Innovation, Science, and Technology (China-Israel collaboration, proposal 005652), the Israel Science Foundation (grants 1535/20, 3062/20, 249/25), and the Volkswagen Foundation (ZN4041). F.J. is supported by the National Key R&D Program of China (2023YFE0113400), the Scientific Research Innovation Capability Support Project for Young Faculty (ZYGXQNJSKYCXNLZCXM-H3), CAMS Innovation Fund for Medical Sciences (2024-I2M-TS-020 and 2023-I2M-2-001), and the Non-profit Central Research Institute Fund of Chinese Academy of Medical Sciences (2023-PT310-04). We thank the Core Facilities and Service Centers, and Dr. Jingdong Song, NIPB, CAMS&PUMC for assistance with ultracentrifugation and electron microscopy work. N.N. would like to thank his family, especially his wife and children for the support and inspiration.

## Author contributions

N.N. and A.L. conceived the idea. N.N. performed all bioinformatic analysis. Z.W. and X.F. performed the experiments. N.N., Z.W., F.J., and A.L. wrote the paper.

## Competing interests

The authors declare no competing interests.
