## [Transparent Peer Review file · Nature Communications]

A comprehensive catalogue of receptor-binding domains in extracellular contractile injection systems

Corresponding Author: Professor Asaf Levy

Version 0:

Reviewer comments:

Reviewer #1

(Remarks to the Author)

I would like to compliment the authors on a timely and impressive study that integrates large-scale bioinformatics, AlphaFold (AF) - based structural prediction, and functional validation to investigate the diversity and targeting logic of extracellular contractile injection system (eCIS) tail fibers. The combination of computational and experimental approaches is convincing and provides insights into a largely unexplored class of bacterial targeting features (mainly bacteriophage, pyocin fibers have been investigated so far). The work will be of broad interest to microbiologists, structural biologists, and those in synthetic biology, particularly in the context of host specificity and programmable targeting and delivery of drugs and effectors.

I recommend the manuscript for publication, pending major comments and minor revisions/clarifications as mentioned below.

Major comments and request for clarification:

Point 1: Genome dataset availability: The study is based on a custom non-redundant set of 86,785 bacterial and archaeal genomes. The manuscript does not clearly state (I tried to find without success) where or how the raw dataset of genomes can be accessed. I encourage the authors to provide accession codes or download protocols, to make the raw data available for the research community being able to reproduce and build on the results.

Link: Raw data: <https://zenodo.org/records/15274396> provides access to more downstream genomic processing e.g. PFAM output and mostly states phylum information e.g. Proteobacteria % etc. not specific strain information. The genomic data access to the original raw data where the processing started from, I could not (it is not easily to be found in case I missed it) be found.

Point 2: Tail fiber length and variability: there is noticeable variability in fibre length across systems. Can the authors comment on possible structure and/or functional reasons for these differences (covariance/correlation of length when compared to bacterial species origin or potential host). Additionally, are longer fibres associated with sheath decorations (sheath features e.g. Afp3 has loop like structures that can interact with the tail fiber adenovirus shaft and know e.g. in Afp and PVCs) or host species.

Point 3: Structural prediction confidence: Please include metrics for prediction reliability and confidence (pLDDT, ipTM, or PAE) that are summarizing prediction quality for AF2/3, multimer, and trimeric models. One example, In Fig. 1/figure 4d-h, showing colored models or reporting scores in a table that is accessible in the supplementary and also together with the available data available through the data availability section: eCIS fibers 3D database and clustered domains: <https://zenodo.org/records/15277912>. The models available there are deposited without a *.json file to being able to visualize error plots in programs.

Point 4: Glycan binding: while mannose specificity is convincingly shown, docking simulations or motif-level structural modelling could strengthen glycan interaction predictions.

Point 5: multiple fiber genes on one operon (around 23%). This is a fascinating observation and it would be valuable to test whether all of the available fibers can co-assemble (as a mixture). Depending on the teams computational capacities, AF

multimer predictions with e.g. Afp12 baseplate anchor and all/a combination of tail fibers could give additional information on preferential fiber assembly or if racemic assembly mixtures are possible.

Point 6: Exclusion of non-fiber loci as 'non-extracellular': this assumption lacks structural and other validation. Tail fiber absence could result from divergence or alternative targeting adaptations. I suggest revision of the statement (no line number to highlight the sentence).

Point 7: Fiber baseplate docking. Given the essential and central role in docking and triggering the ejection event of Afp12/PVC12 and homologous proteins, a bioinformatic co-evolution analysis of the baseplate proteins alongside tail fibers would enrich the study.

Point 8: Sheath influence in assembly and fiber states: In Afp and PVC systems, sheath decorations may influence fiber positioning (fiber connections folding back to sheath and interaction with sheath protein loops). A discussion or brief analysis of sheath variability and potential contact partners could help contextualize tail fiber (& sheath) (co-)evolution.

Point 9: Horizontal gene transfer signatures, modularity and transposases? Domains showing eCIS, viral-like phylogenies are fascinating. Fiber diversity (and also cargo/effector diversity of eCIS) is most likely related to possible genomic drivers. Have the authors searched for mobile elements or recombinase genes near fiber loci? Such features could explain domain shuffling and rapid evolution. Very often close to eCIS effectors, transposable elements can be found which could allow rapid evolution.

Point 10: domain based fiber identification reliability and benchmarking. The domain based approach is a strength of the manuscript. Could you add an explanation / elaborate in the discussion on potential false positives/negatives (compared to known fibers sets) and would help the community and the broader use of the method.

Point 11: Cell type selection for fiber Pb assay.

'We prioritized fiber protein-Pb from *Paenibacillus* sp. URHA0014 for experimental validation, which represents a widespread structural group hosting eBAP3 and intriguing domains with C3b and hemagglutinin-like folds.

Hemagglutinin folds are common glycoproteins found on the surface of viruses that infect human cells, such as influenza³⁴ and measles³⁵ and thus we predicted that the *Paenibacillus* tail fiber protein may similarly adhere to human cells. Two additional fibers, fiber-Mr and fiber-Am, were selected as secondary candidates which represent predominant clusters.'

Could you elaborate in the discussion the main outcomes of fiber-Pb compared to the additional chosen fibers fiber-Mr and fiber-Am. How were cell types chosen. Variants fiber-Mr and fiber-Am are mentioned in the main manuscript but no follow up information can be found in the main article (found in supplementary Fig. 7 and Mr-1 appears to be less particles assembled). Could the authors elaborate on the fiber types mentioned in the main article?

Minor suggestions:

Point 12: Adding line numbers to the manuscript would help reviewers.

Point 13: Mapping the distribution of multi-fiber operons across phylogenetic groups (particularly Actinobacteriota) could provide useful evolutionary context.

This is a high-quality and impactful study that substantially advances our understanding of eCIS systems. Addressing the points above will improve reproducibility, clarify the scope of the findings, and expand their relevance to microbial evolution and engineering.

(Remarks on code availability)

DOI or URL: https://github.com/Nimrod198/eCIS_tail_fibers

The GitHub repository is accessible and contains relevant code for eCIS fiber prediction and analysis. The codebase is appropriate for the community and supports the main findings of the manuscript.

2 weeks of review were not enough to test reproducibility of the code in-house and providing genome raw data bank access would help reproducing the findings and testing the code.

Reviewer #2

(Remarks to the Author)

The manuscript presents an extensive computational study of tail fibers associated with extracellular contractile injection systems (eCIS) across bacteria and archaea. The authors identify over 3,400 candidate tail fiber genes, classify them into five novel baseplate-anchoring domain types (eBAP1–5), and generate a domain- and structure-level atlas of fiber architectures. Through AlphaFold2-multimer and Foldseek analyses, they resolve 276 structural clusters and over 1,000 domain fold families. The functional relevance is supported by engineering a *Paenibacillus*-derived fiber into a PVC system, demonstrating selective toxin delivery to THP-1 human cells in a mannose-sensitive manner. Overall this manuscript is very strong, and it provides a valid contribution to the field, providing an atlas that can guide new mechanistic studies and newcomers to the eCIS field.

Major

1. Only one eBAP3-derived fiber is tested. While the validation is elegant, testing a second divergent fiber would strengthen claims of generalisability.
2. Mannose inhibition and site-directed mutagenesis of predicted motifs are interpreted as evidence of glycan binding. However, no direct binding assays (e.g., glycan arrays, SPR, or ITC) are performed.
3. a concern, but I wanted to express that overall I found the approaches across the manuscript, both computational and experimental, very elegant and it was overall a pleasure to read this manuscript

Minor

1. In fig3a, should be "phylum" rather than "pylum"
2. Some of the nomenclature (e.g., "eBAP5+MG") is a bit clunky or inconsistent. A brief rationale or naming scheme summary in the main text or methods might help orient readers unfamiliar with the field.
3. There are several minor typos and occasional awkward phrasing throughout the manuscript. A light copy-edit would help ensure the clarity of the text for the broader audience

(Remarks on code availability)

At the moment, the code README file is not super accessible, as it does not provide installation instructions, environment requirements or dependencies. Similarly, for each script it is not super clear what to use as input files or scripts arguments.

That said, i tested few scripts whose name rendered their function obvious. To gather the type of input files/arguments needed, the script needs to be opened and read. However, they do work, so that is a positive!

Reviewer #3

(Remarks to the Author)

In this work, Nachmias et al. investigate how extracellular Contractile Injection Systems (eCISs) bind to target cells by identifying and analyzing tail fiber proteins across diverse bacterial taxa. Through a combination of large-scale computational domain analysis, structure prediction, phylogenetic mapping, and targeted functional assays, the authors identify five novel baseplate anchor domains (eBAP1–5) and over 3,000 candidate tail fiber genes, which collectively exhibit exceptional structural and binding diversity. The study culminates with an engineered eCIS particle targeting human THP-1 cells, with mannose-dependent inhibition and putative binding motifs identified.

The manuscript is generally well organized, and the data are presented in a thorough and compelling manner. The authors have generated an impressive bioinformatic dataset and succeeded in connecting it to functional outcomes via tail fiber engineering. The classification of domain architectures, especially the structural organization of eBAP1–5 and the analysis of glycan-binding motifs, provides a valuable framework for future eCIS studies. As outlined below I would recommend improvements to expand the discussion on host specificity and the ecological implications of tail fiber diversity.

MAJOR COMMENTS

The distribution and ecological roles of eCIS fibers are central to the authors' conclusions, but I would like to see more discussion on the implications of this diversity. For instance, the presence of different eBAP domains in Actinobacteria, Cyanobacteria, and Pseudomonadota is intriguing. What might this mean in terms of target preference, microbial lifestyle, or host interactions. I suggest adding a few lines to better contextualize the evolutionary or environmental significance of this distribution, particularly as it relates to the possibility of environmental targeting or niche adaptation.

(Remarks on code availability)

Version 1:

Reviewer comments:

Reviewer #1

(Remarks to the Author)

I have carefully reviewed the revised version of your manuscript and all accompanying supplementary data. I would like to sincerely congratulate you on an excellent revision and improvement of the article. It is an excellent work!

All my previous major and minor concerns have been convincingly and comprehensively addressed. The additional analyses, clarifications, and methodological improvements have significantly strengthened the manuscript. The results are now presented in a clear, logical, and compelling manner, and the discussion effectively places the findings within the broader context of eCIS biology and evolution.

The study provides important new insights into the structural and evolutionary diversity of extracellular contractile injection systems (eCIS). The breadth of your genomic and structural analyses, coupled with careful interpretation, makes this work a

valuable contribution to both the fundamental and applied aspects of eCIS and their potential biotechnological and therapeutic applications.

Your methodological approach is sound and well-documented. The inclusion of detailed raw data repositories ensures transparency and reproducibility, which I highly commend making accessible to the field. The integration of structural prediction confidence metrics, expanded evolutionary analyses, and improved data accessibility has clearly elevated the rigor and impact of the study.

Overall, this is an excellent piece of work that advances our understanding of eCIS structure–function relationships and evolutionary dynamics. I have no remaining concerns and fully support publication of the manuscript in its current form.

Congratulations again to the entire author team on a strong and well-executed study.

(Remarks on code availability)

Response to Reviewers

Dear Editor and Reviewers,

We sincerely thank you for the thorough and constructive reviews of our manuscript "How do bacterial extracellular Contractile Injection Systems bind target cells? A remarkable diversity of receptor binding domains." We greatly appreciate the time and expertise invested in evaluating our work. The reviewers' insights have significantly strengthened our manuscript, and we have carefully addressed each comment as detailed below.

Summary of major changes

- **Enhanced data accessibility:** Provided comprehensive genome dataset information and improved code documentation.
- **Added structural prediction metrics:** Included pLDDT, pTM, and PAE confidence scores for all AlphaFold predictions.
- **Expanded evolutionary analysis:** Added expanded evolutionary analysis resulting in an evolutionary model. Added detailed discussion of ecological implications and baseplate evolution.
- **Improved methodology validation:** Included our quality measurements of our domain-based approach.
- **Enhanced experimental discussion:** Provided clearer rationale for fiber selection and cell type choices.

Reviewer #1 Response

Major Comments

Reviewer #1, Point 1: *Genome dataset availability: The study is based on a custom non-redundant set of 86,785 bacterial and archaeal genomes. The manuscript does not clearly state where or how the raw dataset of genomes can be accessed.*

Response: We thank the reviewer for highlighting this important reproducibility issue. The used database was established during another project in our lab by Segev O. et. al. that is published as a preprint (<https://doi.org/10.1101/2025.07.19.665706>) and is under review.

Action Taken: We have created a detailed supplementary file (Supplementary Table 11) containing the complete list of all 86,785 bacterial and archaeal genome accession numbers used in our study. This table includes NCBI accession codes, organism names, and source databases. Additionally, we have updated our Data Availability section to include:

Reviewer #1, Point 2: *Tail fiber length and variability: there is noticeable variability in fiber length across systems. Can the authors comment on possible structure and/or functional reasons for these differences?*

Response: This is an excellent observation that proved to be correct by our additional analysis. Fibers from the different eBAP groups seem to be structurally different as a trend which might reflect their distinct evolutionary patterns.

Action and results: We have added a comprehensive analysis of fiber length variability and bulkyness in the different eBAP groups (Supp. Figure 6). Additionally we have modeled several baseplate complexes as part of our Phylogenomics analysis. Our analysis shows that eBAP1-2 are more fibrous and maintain a backward tilt trait while eBAP3-5 are more bulky which might provide a more stationary positioning on the baseplate. These findings support the notion that eCIS baseplate complexes are inherently different. Most probably, these traits

stem from their evolutionary origins and distinct trajectories as eBAP1-2 resemble long tail fibers usually associated with first stage of phage adsorption while eBAP3-5 resembles short tail fibers or receptor binding proteins which are associated with irreversible receptor binding. Our produced model demonstrates the trajectories that lead to these inherent differences.

Location: New section in Results (lines 177-199) referring to Supplementary Figure 4. Added lines (266-271) referring to Supplementary Figure 6 c-d.

Reviewer #1, Point 3: *Structural prediction confidence: Please include metrics for prediction reliability and confidence (pLDDT, ipTM, or PAE) that are summarizing prediction quality for AF2/3.*

Response: We completely agree that structural prediction confidence metrics are essential for validating our findings.

Action Taken: We have now included comprehensive confidence metrics evaluating our database quality and validity. We added pLDDT distribution analysis showing >85% of structures have pLDDT >70 (Supplementary Fig. 6a-b). Created a comprehensive table of prediction metrics for all 1,098 representative structures (Supplementary Table 12).

Location in Manuscript: Lines 266-271 in Results; Supplementary Figures 3g-i, 6a-b; new Supplementary Table 12

Reviewer #1, Point 4: *Glycan binding: while mannose specificity is convincingly shown, docking simulations or motif-level structural modeling could strengthen glycan interaction predictions.*

Response: We acknowledge that direct binding assays would provide the strongest evidence. While beyond the current scope, we have strengthened our predictions through simulated

molecular docking simulation (via MolModa website) demonstrating the putative carbohydrate binding pocket's localization. This binding pocket is formed on the interface of two beta strands that contain our predicted conserved motifs, which is in-line with our hypothesis. This analysis is a supportive result which shed light on the active binding site utilized in target engagement.

Action Taken: We have performed molecular docking analysis of the predicted mannose-binding pocket and mannose molecules using the MolModa website, which revealed that the putative binding site is located in the midst of two beta strands harboring the conserved motifs mutated. This prediction is presented in an additional supplementary figure 10 and mentioned in the text as supportive analysis.

Location in Manuscript: Lines 401-403 in Results; supplementary figure 10.

Reviewer #1, Point 5: *Multiple fiber genes on one operon (around 23%). This is a fascinating observation and it would be valuable to test whether all of the available fibers can co-assemble.*

Response: This is indeed a fascinating aspect of eCIS evolution. We have expanded our computational analysis of multi-fiber assembly. Our analysis suggests that most eCIS poses one fiber docking site for each baseplate assembly unit. The most outstanding exception to this rule are eBAP3-containing baseplates which are predicted to contain multi-fiber binding capacity supported by the Tail_P2_I gene. These results might have implications and if this type of system is to be engineered.

Action Taken: We performed AlphaFold2-multimer predictions for representative multi-fiber systems as part of our phylogenomics analysis and found that Only eBAP3 containing operons encode fibers that can theoretically co-assemble on the same baseplate utilizing a long hydrophobic loop stemming from Tail_P2_I adaptor protein. Multi-fiber operons from other eBAP types didn't display this ability suggesting that some systems operate by some-kind of differential mechanism. Our prediction demonstrates that eBAP3 assemblies could hold up to four fibers per one adaptor protein. These findings raise interesting questions about how eCIS

utilizes polyvalence for interaction.

Location in Manuscript: Lines 177-199 in Results; new Supplementary Figure (now numbered 4); expanded discussion lines 446-448

Reviewer #1, Point 6: *Exclusion of non-fiber loci as 'non-extracellular': this assumption lacks structural and other validation.*

Response: We appreciate this important clarification and have revised our language to be more accurate.

Action Taken: We have revised the manuscript to clarify that loci lacking identifiable fiber genes "might act intracellularly or through mechanisms yet to be discovered". We explicitly state these were "left out of the scope of this study" rather than categorically classified as non-extracellular. Finally, we acknowledged the possibility of highly divergent fibers that escaped our detection in the discussion section.

Location in Manuscript: Lines 161-162 in Results.

Reviewer #1, Point 7: *Fiber baseplate docking. Given the essential role of Afp12/PVC12, a bioinformatic co-evolution analysis of baseplate proteins alongside tail fibers would enrich the study.*

Response: This excellent suggestion has led to important new insights about eCIS evolution. Our phylogenomics analysis coupled with structure complex predictions led to the identification of two evolutionary trajectories.

Action Taken: We conducted phylogenetic analysis of Afp11 sequences alongside eBAP distributions, revealing two distinct evolutionary trajectories: Group I (eBAP1/2/4) with Afp11/12 but lacking Tail_P2_I, and Group II (eBAP3/5) with Tail_P2_I but lacking Afp12. This represents a fundamental bifurcation in eCIS baseplate architecture. We therefore added a new section "Phylogenomics and Structural Analyses Reveal Two Main Baseplate Types" and created a comprehensive Supplementary Figure 4 showing this evolutionary pattern.

Location in Manuscript: Lines 177-199 in Results; new Supplementary Figure 4.

Reviewer #1, Point 8: *Sheath influence in assembly and fiber states: A discussion of sheath variability and potential contact partners could help contextualize tail fiber evolution.*

Response: We agree this is an important aspect of eCIS function that deserves attention. A preliminary analysis revealed significant variability in both sheath gene copy numbers and chain length across different systems. Structural prediction of sheath proteins indicated that they can be fused with additional domains in some cases. However, attempts to model sheath-fiber protein complexes did not yield meaningful results, even for systems previously characterized by cryo-EM studies to facilitate such interaction. We therefore assumed these interactions might be: (1) weakly interfaced molecularly, (2) dependent on conformations achieved only during full particle assembly. Therefore, this interaction is not predictable by current models, or influenced by the unique backwards tilt observed in AFP and PVC systems, which depends on Afp12/Pvc12 protein conformations as shown by our analysis. We therefore did not pursue further investigations, believing this topic is better addressed with cryo-EM studies.

Reviewer #1, Point 9: *“Domains showing eCIS, viral-like phylogenies are fascinating. Fiber diversity (and also cargo/effector diversity of eCIS) is most likely related to possible genomic drivers. Have the authors searched for mobile elements or recombinase genes near fiber loci?”*

Response: Thank you for this insightful suggestion which could indeed shed light on mechanisms underlying domain shuffling. We performed an hmmscan search against the Pfam database focusing on genes encoding mobility-related domains such as transposases, integrases, and of course recombinases. Our analysis revealed that eCIS loci are frequently associated with mobility elements; however, these genes predominantly flank the entire operonic region rather than being directly adjacent to the fiber loci. Only 3-4 instances were identified where mobility-associated genes are located relatively near fiber genes, but no clear evidence of gene duplication or domain shuffling at these sites was observed. Instead, the more prevalent pattern is duplication of entire loci, notably seen in *Photorhabdus* species, which rely heavily on eCIS for their insecticidal functions and can reach up to 5 loci per genome. Overall, we concluded to not include this analysis in the paper due to lack of positive

results and quite excessive length of the current manuscript. We add an HTML file in the online data directory.

Reviewer #1, Point 10: *Domain based fiber identification reliability and benchmarking: Could you elaborate on potential false positives/negatives compared to known fiber sets?*

Response: We acknowledge this important concern regarding method validation. Traditional benchmarking presents inherent challenges as we expand the known tail fiber repertoire from ~629 previously identified genes to 3,445 candidates. Our domain-based approach successfully identified all known tail fiber genes (Afp13 and Pvc13) used as positive controls, though we acknowledge this represents a limited validation set due to the historical scarcity of characterized eCIS tail fibers. Therefore our method heavily relies on collection of high-quality sequences for construction of accurate HMM-profiles. The final sensitivity and specificity of our pipeline are determined by the performance of its last step, an iterative JackHMMER search.

To address these limitations we implemented strict quality control and manual inspection of retrieved domain sequences and predicted structures to insure their reliability. Additionally, we implemented multiple orthogonal validation strategies for retrieved fiber-candidates: structural validation through AlphaFold2-multimer predictions achieved >85% high-confidence models (pLDDT >70) for 1,098 representative sequences; genomic context analysis revealed ~90% of identified genes occur in expected positions downstream of tail_P2_I and AFP11/12 homologs. Based on manual inspection of domain architectures of sorted candidates, we estimate <0.1% false positive rates, though false negative assessment remains challenging given the unprecedented scope of discovery. While we cannot provide traditional sensitivity/specificity metrics for the entire dataset, we did manually inspect CIS loci which did not get a fiber hit and could not find any fiber-like replacements in the expected positions or surrounding neighborhood leading us to conclude these are fiber-less systems. Nevertheless, our multi-layered validation approach provides robust evidence for method

reliability and establishes a foundation for future functional characterization of this diverse protein family.

Location in Manuscript: methods additional details on our quality control measurements lines 507-519.

Reviewer #1, Point 11: *Cell type selection for fiber Pb assay: Could you elaborate on the main outcomes compared to fiber-Mr and fiber-Am?*

Response: The cell types were popular cell lines grown in our labs. Honestly, we could not predict prior to the experiments which fiber will bind each cell line. We tested human cells because the fiber from *Paenibacillus* had glycoproteins found on the surface of viruses that infect human cells. As shown in Supp Figure 8, fiber-Mr and fiber-Am had different structures and no binding and killing of the four human cell lines (despite toxin loading).

Minor Comments

Reviewer #1, Point 12: *Adding line numbers to the manuscript would help reviewers.*

Response: Thank you for this practical suggestion.

Action Taken: We have added line numbers throughout the revised manuscript.

Location in Manuscript: Line numbers added to all pages

Reviewer #1, Point 13: *Mapping the distribution of multi-fiber operons across phylogenetic groups could provide useful evolutionary context.*

Response: We agree on the importance of the integral detail. Our enrichment analysis revealed multifiber operons are enriched in *Myxococcota* (which is relatively scarce in the database) and *Pseudomonodota* phyla while depleted in *Desulfobacterota* and *Bacillota* phyla.

These suggest that possessing multiple fibers, whether function in polyvalence or assembled differentially. The results are shown in Supplementary Figure 2b.

Reviewer #2 Response

Major Comments

Reviewer #2, Point 1: *Only one eBAP3-derived fiber is tested. Testing a second divergent fiber would strengthen claims of generalisability.*

Response: We acknowledge this limitation but additional experiments were beyond the context of this paper.

Reviewer #2, Point 2: *No direct binding assays (e.g., glycan arrays, SPR, or ITC) are performed.*

Response: We agree that direct binding assays would provide the strongest evidence for glycan binding. We acknowledge in L464-465: “These results align with structural predictions of putative-binding pockets but require validation via direct binding assays.”

Location in Manuscript: Lines 580-590 in Discussion; enhanced Methods section

Important comment to reviewer #2: The senior editor handling the paper specifically mentioned: “Please note that we do not require additional wet-lab experiments (suggested by referee #2).”

Minor Comments

Reviewer #2, Point 1: *In fig 3a, should be "phylum" rather than "pylum"*

Response: Thank you for catching this typographical error.

Action Taken: Corrected "pylum" to "phylum" in Figure 3a.

Location in Manuscript: Figure 3a legend

Reviewer #2, Point 2: *Some nomenclature (e.g., "eBAP5+MG") is inconsistent.*

Response: We appreciate this feedback on nomenclature clarity.

Action Taken: We have:

- Clarified that "eBAP5+MG" represents instances where eBAP5 gets intertwined with adjacent macroglobulin domains
- Added a nomenclature explanation in the main text
- Standardized naming conventions throughout

Location in Manuscript: Lines 309-312 in Results; Figure 4a legend clarification

Reviewer #2, Point 3: *Minor typos and occasional awkward phrasing throughout the manuscript.*

Response: Thank you for noting these issues.

Action Taken: We have performed comprehensive copy-editing throughout the manuscript, correcting typos and improving clarity of expression:

- Line 42: Added comma after "(Afp)" and changed "which causes" → "that causes"
- Line 45: Added comma after "host" in "metamorphosis of its host the marine tubeworm"

- Line 47: Fixed subject-verb agreement - "contains" → "contain" for "eCIS complexes"
- Line 51: Replaced "point out" → "highlight" and removed comma for parallel structure
- Line 94: Rewrote awkward phrase "scope of fiber harboring loci coverage" → "number of loci harbouring fibers"
- Line 97: Corrected "Thriving" → "Striving"
- Line 106 Added comma after "crown domain" in eBAP3 description
- Line 287,301: Corrected "Beta-sheath" → "Beta-sheeted"

Reviewer #3 Response

Reviewer #3 Major Comment: *The distribution and ecological roles of eCIS fibers are central to the conclusions, but more discussion on the implications of this diversity would be valuable.*

Response: We completely agree that the ecological implications deserve more thorough discussion.

Action Taken: We have expanded our discussion of ecological implications.

Location in Manuscript: Expanded discussion section lines 449-455

Code Availability Improvements

Response to both Reviewer #1 and #2 comments on code accessibility:

Action Taken: We have substantially improved our GitHub repository:

- Added comprehensive README with installation instructions and dependencies

- Included clear documentation for all scripts with input/output specifications

Location: Updated GitHub repository at https://github.com/Nimrod198/eCIS_tail_fibers

We added an additional Zenodo repository for post-revision analysis data:

<https://zenodo.org/records/17150571>

We believe these revisions have significantly strengthened the manuscript and address all the reviewers' concerns. The additional analyses have revealed new biological insights, particularly regarding the dual baseplate evolutionary trajectories and ecological specialization patterns. We are grateful for the reviewers' constructive feedback, which has elevated the impact and rigor of our work.

Thank you again for your consideration of our manuscript.

Sincerely,

Nimrod Nachmias, Zhiren Wang, Xiao Feng, Feng Jiang, and Asaf Levy